

# Enhancing neural collaborative filtering using hybrid feature selection for recommendation

Baboucarr Drammeh[1,2] and Hui Li[1]

[1] College of Computer Science and Technology, Guizhou University, Guiyang, Guizhou, China
[2] School of Information Technology and Communication, University of the Gambia (UTG), Banjul, Kanifing, Gambia

## ABSTRACT

The past decade has seen substantial growth in online transactions. Accordingly, many professionals and researchers utilize deep learning models to design and develop recommender systems to suit the needs of online personal services. These systems can model the interactions between users and items. However, existing approaches focus on either modeling global or local item correlation and rarely consider both cases, thus failing to represent user-item correlation very well. Therefore, this article proposes a deep collaborative recommendation system based on a convolutional neural network with an outer product matrix and a hybrid feature selection module to capture local and global higher-order interaction between users and items. Moreover, we incorporated the weights of generalized matrix factorization to optimize the overall network performance and prevent overfitting. Finally, we conducted extensive experiments on two real-world datasets with different sparsity to confirm that our proposed approach outperforms the baseline methods we have used in the experiment.

## INTRODUCTION

The large amounts of information generated by online services have recently challenged online users to identify meaningful recommendations. However, recommendation systems can help users find valuable information faster, and they are frequently utilized in services such as e-commerce (*Chen et al., 2016*), social media recommendations (*Guy et al., 2010*), and online video services (*Covington, Adams & Sargin, 2016*).

Traditional approaches provide recommendations based on the similarities between users and items, which can be categorized into collaborative filtering, content-based, and hybrid recommendation systems (*Lu et al., 2015*). Collaborative filtering has been researched and heavily used in systems based on personal recommendation. It utilizes the possibility of items users may be interested in based on their historical interactions (*Breese, Heckerman & Kadie, 1998*). Furthermore, content-based recommendations uses additional features about users or items to recommend similar items (*Çano & Morisio, 2017*; *Javed*

Corresponding author
Hui Li, cse.HuiLi@gzu.edu.gm

*et al., 2021*). Finally, the hybrid method combines two or more recommendation techniques (*Çano & Morisio, 2017*).

Most traditional recommendation systems are limited when a large amount of data is to be analyzed (*Gasmi, Bouhadada & Benmachiche, 2020*), Moreover, they rely on a linear kernel that does not fully represent user-item interaction. Accordingly, researchers have recently adopted deep learning-based approaches to develop recommender systems (*Pan, He & Yu, 2020*) since they can learn complex nonlinear relationships and handle various data types. Several deep learning-based methods have been proposed due to their ability to deal with high-dimensional features effectively. Collaborative deep learning methods based on matrix factorization are known to provide satisfactory performance (*He et al., 2017*; *Zhang et al., 2016*). Matrix factorization represents a given user and item as an embedding and learns their relationship using an inner product between the user and item embeddings. Despite its effectiveness, matrix factorization is limited since it uses an inner product as the interaction function (*He et al., 2018*), which assumes that the embedding dimensions are independent of each other and perform a multiplication between them, thus limiting the expressiveness of the model.

In this study, we proposed a recommender algorithm using a hybrid feature selection module (HSFM) to capture the useful global and local high-dimensional relationship between users and items. Our proposed approach utilizes convolution to capture the valuable nonlinear relationship between users and items by learning the outer product matrix. To verify the effectiveness of our model, we conducted experiments on two real-world datasets, MovieLens and Pinterest. Experimental results demonstrate that our proposed model outperforms the baseline methods significantly. The main contribution of the article is summarized as follows:

1) The stack interaction map is introduced to increase the input features expressiveness and allow the interaction map to encode more latent signals.
2) To effectively capture the correlations between items, we also leverage a hybrid feature selection module, which uses pointwise convolution and general average pooling to learn both local and global item correlations.
3) We also incorporate generalized matrix factorization (GMF) to constrain the network's weight which optimizes the network performance and prevents overfitting.
4) We conducted an extensive experiment on two publicly available datasets to demonstrate the effectiveness of our proposed model.

The remainder of the article is structured as follows. First, the "Related Works" section reviews related works, and "Our proposed methods" elaborates on our proposed method. Then, the experimental results on the two datasets are reported in the "Experiments" section. Finally, we conclude this article in the "Conclusions".

## RELATED WORKS

In this section, traditional recommender systems were examined on how they model the similarity between users and items; second, we also look at deep learning techniques due to

their high-quality recommendation performance and better ability to learn the relationship between users and items.

## Traditional recommender systems

Many recommender algorithms are based on collaborative filtering (*Adomavicius & Tuzhilin, 2005*), which depends on users' past behavior to make predictions. Collaborative filtering-based recommendations are divided into latent factor methods (*Koren, Bell & Volinsky, 2009*) and neighborhood-based methods (*Sarwar et al., 2001*). Neighborhood-based approaches utilize ratings directly to evaluate new items for users. Such models are based on the similarities between a user to a user or item to an item. The similarity between two items is measured as the probability of users rating those items' similarly, which is usually based on the Pearson correlation. On the other hand, the latent factor models users and items as vectors using the same latent space by reducing the number of hidden factors. Latent factors compare users and items directly, where a user's rating of an item is predicted using the inner product between related latent vectors.

Singular value decomposition (*Koren, Bell & Volinsky, 2009*) reduces the number of user-item features to a product of two low-rank matrices. However, one of its drawbacks is the high cost of locating singular value decomposition. Another approach that enhanced the singular value decomposition, SVD++ (*Koren, 2008*), uses implicit and explicit feedback to provide a recommendation and demonstrate improved performance over many matrix factorization models.

Generally, traditional recommender algorithms use a linear kernel that does not better represent the user-item relationship.

## Deep learning base recommender system

Deep learning has developed extensively in the past decade and has been implemented in various fields, such as computer vision, speech recognition, and natural language processing (*Zhang, Yu & He, 2018*). Deep learning learns features directly from data and performs feature engineering automatically, and it has been studied extensively in recommendation systems. The deep learning-based models have demonstrated significant performance over the traditional recommendation system (*Singhal, Sinha & Pant, 2017*). For instance, Neural collaborative filtering (*He et al., 2017*) utilizes a multilayer perceptron to model the interaction function as it represents users and items as a low-dimensional vector in latent space.

In another research, deep matrix factorization (DMF) (*Xue et al., 2017*) utilizes a matrix factorization and a neural network architecture, which uses explicit scores of users and non-preference implicit feedback of items.

The correlation denoising autoencoder (*Pan, He & Yu, 2020*) considers the correlation between users with diverse roles to learn a more robust representation from sparse ratings and social networks. It uses three autoencoders to learn user features taking them as a separate matrix of rating, truster, and trustee. The authors *Liu et al. (2020)* couple deep neural networks with matrix factorization and learn the deep global and local item

relationship of item content by coupling autoencoder with matrix factorization to join the rating and item content information.

Convolutional neural networks (*LeCun et al., 1998*) are prevalent in image recognition, and they are generally made up of a convolution layer, pooling, and a fully connected layer. Convolutions are also used in recommender systems to model the interaction map. For instance, the convolutional factorization machine (*Xin et al., 2019*) is a recommender model that is context aware; it uses self-attention, an embedding layer, and a pooling layer. The authors also used an outer product interaction cube coupled with a 3D convolutional neural network to extract higher-order signals. In another research, the authors of ConvNCF (*He et al., 2018*) also utilize an outer product to explicitly model pairwise correlation instead of just concatenating or mere element-wise multiplication of the embedding. In addition, they also use a convolutional neural network above the interaction map to learn higher-order correlations.

Convolutional factorization machine and ConvNCF use regular convolution, which helps learn local features and does not learn global features well.

Our proposed hybrid feature selection uses deep neural networks to learn both local and global item correlation between users and items. In addition, we incorporate GMF into the model to optimize the overall model performance and prevent overfitting.

## OUR PROPOSED METHODS

### Input and embedding layer

Given a user $u$ and item $i$, $V_u^U$ and $V_i^I$ represent the feature vectors of U and I respectively, and their embeddings can be represented as:

$$p_u = P^T v_u^U, \quad q_i = Q^T v_i^I \tag{1}$$

where $P \in R^{M \times K}$ and $Q \in R^{N \times K}$ represent the embedding matrix for the user and item features, M, N, and K represent the number of users, the number of items, and the embedding size, respectively.

### Interaction map

The outer product was utilized to generate the interaction map since it can learn more information between latent features. For example, the outer product between a user and an item can be defined as:

$$m^t \otimes n^T = m^t n = \begin{pmatrix} m_{d_1} n_{d_1} & \cdots & m_{d_1} n_{d_k} \\ \vdots & \ddots & \vdots \\ m_{d_k} n_{d_1} & \cdots & m_{d_k} n_{d_k} \end{pmatrix} \tag{2}$$

where m and n represent row vectors and denote K-dimensional latent vectors.

If $p_u = m^t$ and $q_i = n^T$ then $p_u$ and $q_i$ are used to obtain the interaction map, and it can be represented as:

$$E(p_u, q_i) = p_u \otimes q_i = p_u q_i^T \tag{3}$$

where $E(p_u, q_i)$ represents a $K \times K$ matrix.

Matrix factorization is not robust for modeling user-item correlation because it considers only diagonal elements and performs simple concatenation. However, the outer product is more robust and encodes more latent signals. The interaction map obtained by the outer product has one pair of latent factors, which may not perform well in the 2D convolution. As a result, we stack the interaction map into a k number of features concatenated along the dimensions. The latent signal of the interaction map determines the k-dimension features passed as input into the 2D convolution. Therefore, the increase in latent features makes the interaction map encode more relational signals, thus making it more expressive. Furthermore, the higher the number of k, the feature dimension of the input to the convolution also increases, making the model more memory and computationally intensive. However, k values greater than three do not guarantee an increase in the accuracy of the model and occasionally lead to overfitting.

## Convolution module

The stack interaction map that encodes richer latent features is used as the first input to the convolution module. The convolution module is a three-layer convolution that learns the local features between users and items. The first convolution is the input layer, followed by the two hidden layers, which help to learn more meaningful information between the users and items. The input convolution layer takes the input channel of the stack interaction map, an output channel size of 32, and a kernel size of 2, while the hidden layer utilizes input and output channels of 32 with a kernel size of 2. The convolution layer is mathematically represented as:

$$f_l^k(a, b) = \sum_c \sum_{p_u, q_i} E_c(p_u, q_i) * e_l^k(u, v) \tag{4}$$

where $*$ is the convolution, $f_l^k(a, b)$ represent the element of the feature matrix, $E_c(p_u, q_i)$ represents the element of the input stack interaction map $E_c$ of channel **c**, which is element-wise multiplied by $e_l^k(u, v)$ index of the $k$th convolution kernel $k_l$ of the $l$th layer. The feature map of the k[th] convolutional operation can be expressed as:

$$F_l^k = [f_l^k(1, 1), .., f_l^k(a, b), .., f_l^k(A, B)] \tag{5}$$

$F_l^k$ represent the input feature matrix for the $l$th layers and $k$th neuron, A *and B* represent the total number of rows and columns of the feature matrix, respectively.

A convolution block or network can obtain multiple convolutional layers, dropout, and activation map for extracting meaningful information and adding non-linearity to learned complex patterns. In addition, a dropout layer is introduced to reduce overfitting, which negatively impacts the model prediction.

## Hybrid feature selection module

The convolution module with a stacked interaction map which is used as input to our model, is prone to overfitting. However, it can capture relational representation well. The convolutional module introduced in "Convolution Module" is followed by a dropout layer and ReLU (*Agarap, 2018*) activation function. Nevertheless, the dropout minimizes

overfitting at the expense of removing valuable feature relational representation. Therefore, we suggest the hybrid feature selection module (HSFM) to bridge the gap between overfitting and losing valuable information. The HSFM module takes in two inputs, **y**, and **x**, representing the output before dropout and after the ReLU activation function.

The HFSM aggregates two distinctive features to complement each other. A convenient approach to aligning two distinct feature relationships is to learn their local relationship, which can be obtained using pointwise convolutions. Therefore, the inputs **x** and **y** are summed and passed to two branches. The first branch accesses the global feature using general average pooling (GAP), and the second focuses on the local feature relationship. The two branches have two pointwise convolutions, each followed by binary normalization that minimizes the feature variation and a ReLU non-linearity. Finally, the HFSM combines global and local relationships by applying sigmoid activation on the sum of the two-branch feature, which is expressed as:

$$out = x \otimes s(m) + y \otimes (1 - s(m)) \tag{6}$$

where $s$ represents the sigmoid function and **m** is the summation of the global and local branches.

Generally, hybrid feature selection modules access the global and local relationships from the two distinct branches that complement each other to extract better feature representation without introducing overfitting.

## Generalized matrix factorization

GMF learns from data without uniform constraints and is more expressive than linear Matrix Factorization. Therefore, we combine the losses of GMF (*He et al., 2017*) and our proposed model to update the overall model weight, which obtains a better result and further avoids overfitting. GMF uses an element-wise product of a latent vector of users $p_u$ and items $q_i$, which can be represented as:

$$z^{gmf} = \varphi_1(p_u, q_i) = p_u \otimes q_i \tag{7}$$

The prediction of GMF is also represented as:

$$\grave{p}_{ui} = \sigma\left[h^T z^{gmf}\right] \tag{8}$$

where $\sigma$ represents the sigmoid given as $\sigma(a) = 1/(1 + e^{-a})$ and h is the weight of the output layer.

## Fusion of our proposed method and GMF

Fusion in a convolutional network joins two or more features using an element-wise product, element-wise summation, or concatenation. Concatenation provides a better representation of latent features at the expense of computational complexity and memory consumption. Alternatively, features can be fused by combining the losses of different network modules, constraining the model's overall weight by considering the submodules' special functions.

The proposed network uses Bayesian personalized ranking (BPR), a pairwise loss function, since it measures the dependency between data points and can measure the complex relationship between data points. It can be represented as follows:

$$\mathcal{L}1 = \sum_{u=1}^{N} \sum_{i \in I_u^+, j \in I_u^-} -log\sigma(\hat{x}_{uij}) + \lambda\Omega(\Phi) \tag{9}$$

where $\hat{x}_{uij} = p_u^T q_i - p_u^T q_j$, $\sigma(x) = 1/(1 + exp(-x))$ is the sigmoid function and $\lambda\Omega(\Phi)$ is the regularization.

On the other hand, the GMF model utilized the log loss function, which is a pointwise loss function, and it is easily computable and differentiable by the optimizer. Pointwise loss is also more flexible since it can be applied in many applications and is robust to outliers and noise in data. It can be expressed as:

$$\mathcal{L}2 = - \sum_{(u,i) \in \mathcal{R}^+ \mathcal{R}^-} \left[ f_{ui} \log f_{ui}) + (1 - f_{ui}) \log\left(1 - \hat{f}_{ui}\right) \right] \tag{10}$$

where $R^+$ represents positive a training instance, $R^-$ represents the set of negative training instances and $f_{ui}, \hat{f}_{ui}$ is the prediction and label of the GMF.

$$Loss = \alpha * L1 + \beta * L2 \tag{11}$$

where $L1$ is the BPR loss, $L2$ is the log loss for the GMF, and $\alpha$ and $\beta$ are the weighted coefficients.

The $\mathcal{L}oss$ combined $\mathcal{L}1$, and $\mathcal{L}2$ to constrain the weight of the overall proposed method, essentially avoiding overfitting and further improving the recommendation performance. In addition, we added the weighted co-efficient $\boldsymbol{\alpha}$ and $\boldsymbol{\beta}$ values 0.5 and 0.75, respectively, to tune the impact of the sub-networks losses for the ease of model minimization as the training epochs increase.

### Final prediction layer

The output of the HFSM is reshaped and flattened using a fully connected layer to facilitate the output prediction. Finally, the result is passed into a sigmoid function to calculate the final prediction score. The array of scores $\hat{f}_{ui}$ and $\hat{y}_{ui}$ represent the prediction scores of GMF and the proposed CNN-based model, respectively.

## EXPERIMENTS

The subsequent section presents our experiments on two publicly available datasets to answer the following questions:

**RQ1** Does the proposed model outperform the baselines in top k recommendations?

**RQ2** Is the proposed stacking of the interaction map helpful for learning from user-item interaction and improving recommendations?

**RQ3** How do key hyperparameter settings influence the performance of our model?

## Experimental settings

### Datasets

MovieLens 1M is a movie rating dataset that contains around 1 million ratings of around 3,900 movies by 6,040 users in which there are five-grade ratings, and each user rated at least 20 items. It is a widely used data set for evaluating recommendation performance.

**Pinterest** is an implicit feedback dataset constructed by *Geng et al. (2015)* for evaluating content-based image recommendations. It has 55,187 users and 9,916 items. The original dataset is sparse, but the preprocessed contains at least 20 interactions. Each interaction represents if a user has pinned an image to their board.

### Evaluation protocols

We use leave-one-out evaluation, a popular method for testing the quality of the ranking for the recommendation. For each user, 256 unrated items are used as test data. We used the hit ratio (HR) and the normalized discounted cumulative gain (NDCG) as the evaluation matrix. Both metrics were calculated for each user, and the average scores were reported.

The hit ratio represents the relevant time items in the top-n list of an individual user that appear. It can be represented as:

$$HR@n = \frac{hits}{n} \tag{12}$$

where n represents the number of top n items generated from the methods, a higher value denotes better performance.

NDCG is sensitive to the relevance of higher-ranked items and assigns higher scores to the correct recommendations at a higher rank in the list. NDCG is defined as follows:

$$nDCG_p = \frac{DCG_p}{IDCG_P} \tag{13}$$

where $IDCG_P = \sum_{i=1}^{|REL_P|} \frac{2^{rel_i - 1}}{\log_2(i+1)}$

$\Re L_p$ = list of useful items and p = position

$$RMSE = \sqrt{\sum_{(u,i) \in \mathcal{R}_{test}} \frac{(r_{u,i} - \hat{r}_{u,i})^2}{|\mathcal{R}_{test}|}} \tag{14}$$

$$MAE = \frac{1}{|\mathcal{R}_{test}|} \sum_{(u,i) \in \mathcal{R}_{test}} |r_{u,i} - \hat{r}_{u,i}| \tag{15}$$

where $r_{u,i}$ represents the actual rating, $\hat{r}_{u,i}$ represents the prediction and $\mathcal{R}_{test}$ represents the number of ratings in the test set.

### Baselines

To justify the effectiveness of our proposed model, we compare it with the following baselines:

- **MLP** (*He et al., 2017*) is a neural collaborative filtering approach that reduces the matrix of users and items into two submatrices and multiplies them together to learn the interaction function.
- **GMF** (*He et al., 2017*) uses a scalar product to model the interaction between users and items by reducing their metrics into two summaries.
- **DMF** (*Xue et al., 2017*) uses matrix factorization coupled with neural network architecture. It also projects users and items into lower-dimensional vectors in latent space.
- **NeuMF** (*He et al., 2017*) is an item recommendation method that joins hidden layers of GMF and MLP to model the user-item interaction function.
- **ONCF** (*He et al., 2018*) uses a convolutional neural network with an outer product to model the correlation of user-item correlation; it is an improvement of Matrix Factorization.
- **SDMR** (*Tran et al., 2019*) utilized deep learning to learn the signed distance between users and items and produce a recommendation based on the learned signed distance. Specifically, signed distance measures the difference or similarity between two items. SDMR combines two signed distance scores internally: signed-distance base perceptron (SDP) and signed distance base memory network (SDM).
- **CoCNN** (*Chen, Ma & Zhou, 2022*) CoCNN Joins a co-occurrence pattern and Convolutional Neural network to collaborative filtering with implicit feedback. The authors also designed an embedding structure to capture the link between user-item and item-item. They also proposed a multi-task neural network to share the knowledge of the two tasks.

### *Parameter settings*

We implemented our proposed model using Pytorch on Nvidia GTX 1080. All models were optimized using Mini-batch Adagrad, and the learning rate is searched between [0.001, 0.0001, 0.00001, 0.000001, 0.00000001]. The batch size is 256, and the embedding size is 64. ONCF and our proposed model used a channel size of 32. We also use a dropout of 0.2 for our CNN-Based and 0.5 for our CNN-Based+HFSM and CNN-Based +HFSM +GMF model.

## Performance comparison (RQ1)

Table 1 shows a comparison between our proposed model and the baselines that we used in the experiment. The performance evaluation used for the comparison utilized HR@10 and NDCG@10. In addition, for a fair comparison, we trained all the baseline models using BPR loss.

1) Table 1 shows that ONCF has outperformed MLP, GMF, NeuMF, DMF, and SDMR by both HR@10 and NDCG@10 on both datasets.
2) The CNN based on our proposed model has outperformed ONCF by a significant margin in both datasets since it does not lose much information during feature

**Table 1 Different models when generating top-k recommendations on two datasets.**

|  | MovieLens | | Pinterest | |
| --- | --- | --- | --- | --- |
|  | HR@K | NDCG@K | HR@K | NDCG@K |
| MLP | 0.3474 | 0.1997 | 0.1874 | 0.0932 |
| GMF | 0.4680 | 0.2633 | 0.2728 | 0.1402 |
| NeuMF | 0.301 | 0.2159 | 0.2267 | 0.1167 |
| SDMR | 0.2397 | 0.1203 | 0.1148 | 0.0537 |
| DMF | 0.2051 | 0.1143 | 0.2057 | 0.0787 |
| ONCF | 0.3874 | 0.2004 | 0.2780 | 0.1350 |
| Cocnn | 0.5796 | 0.3288 | – | – |
| **CNN-Based** | **0.9305** | **0.3314** | **0.9301** | **0.3313** |
| **CNN-Based+HSFM** | **0.9589** | **0.3416** | **0.9472** | **0.3374** |
| **CNN-Based +HSFM +GMF (ours)** | **0.9733** | **0.3467** | **0.9691** | **0.3452** |

**Note:**
k ∈ {10}. The boldface denotes the persistently increased scores of our proposed networks.

extraction to establish local relationships. In addition, The CNN-based approach utilizes stacking of the interaction map to encode better latent signals and establish a better user-item relationship.

3) The CNN-based approach of our proposed method does not capture the global relationship in the interaction map. Therefore, we introduced the HSFM module for the model to learn global relationships in addition to local ones. This mechanism has also improved performance over the CNN-based in HR and NDCG scores, respectively.

4) The fusion of GMF and our proposed method constrain the overall model weight, thus allowing the model to benefit from sub-networks. This combination obtained the best performance in our proposed methods on both datasets.

5) Since all the networks are trained on the BPRLoss, our proposed model performed better than the baselines on both datasets. Moreover, the proposed method obtained a remarkable performance on not only the CNN-based global and the local feature interaction for high relational modeling but the hybrid of the GMF sub-network, which shows promising results and further constrains the model weight. However, these benefits come at the expense of computation complexity.

From the charts in Fig. 1, our best-proposed model has significantly outperformed the baselines on both datasets at HR and NDCG evaluation metrics. Furthermore, among the baselines, NeuMF outperformed MLP, but it is entirely defeated by GMF, demonstrating that GMF is a simply designed yet powerful prediction model. On the other hand, NeuMF does not achieve the desired result, which may result from the selected optimizer or the poor performance of the underlying MLP in the sparse datasets.

To further evaluate the efficiency of our model, we use RMSE, MAE, and BPRLOSS and compare them to the baselines. Table 2 shows that our proposed model has effectively reduced the loss more than the baselines, thus indicating a strong minimization ability during the model training.
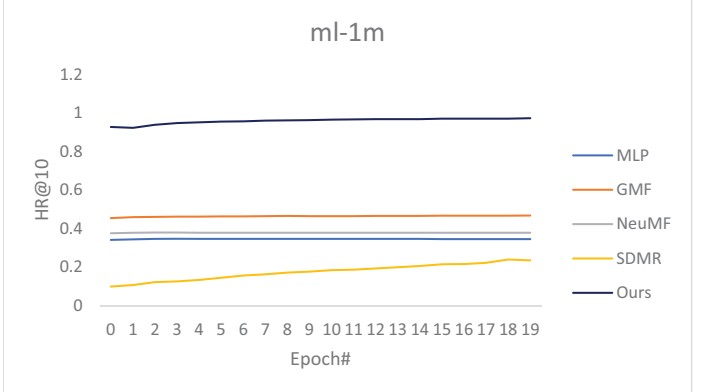 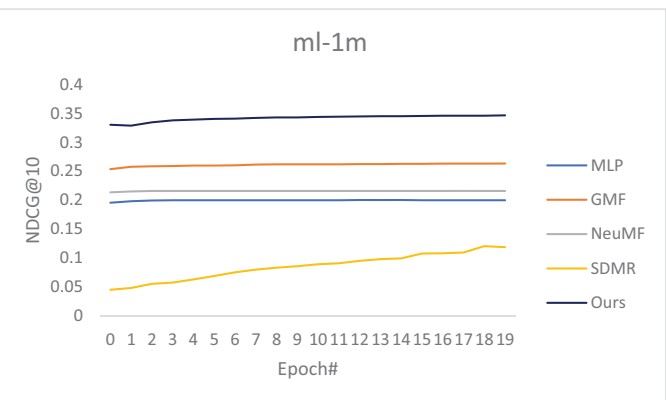

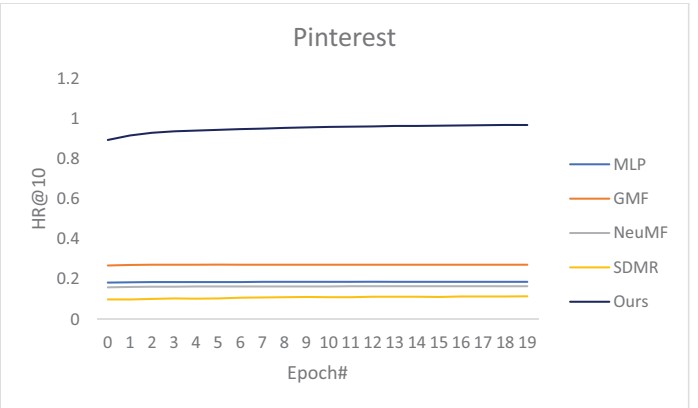 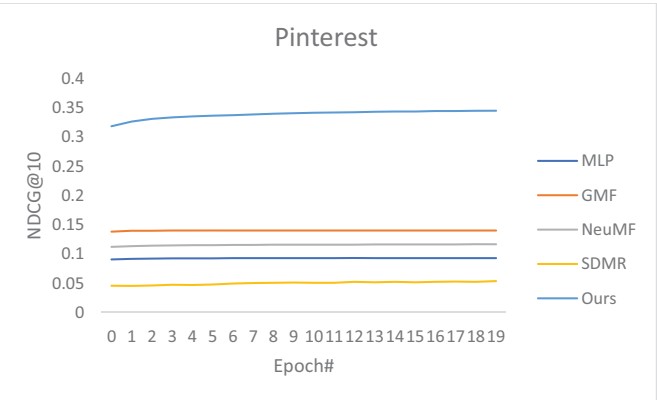

**Figure 1 The graphical representation of the performance of our best-proposed method compared to the baselines using HR@10 and NDCG@10.** (A) HR on MovieLens. (B) NDCG on MovieLens. (C) HR on Pinterest. (D) NDCG on Pinterest. The performance on the Movie-Lens (A and B) and Pinterest (C and D) datasets, respectively.

**Table 2 The performance of our proposed model against the baselines on RMSE, MAE, and BPR loss.**

|  | ML-1M | | |
| --- | --- | --- | --- |
|  | RMSE | MAE | BPRLOSS |
| MLP | 0.3218 | 0.1543 | 0.1281 |
| GMF | 0.327 | 0.1999 | 0.8028 |
| NeuMF | 0.3181 | 0.1533 | 0.1321 |
| SDMR | 8.705 | 7.6734 | 0.6933 |
| DMF | 0.2565 | 0.1324 | – |
| ONCF | 0.2818 | 0.2358 | 0.0158 |
| Cocnn | 0.5413 | 0.3666 | 0.0861 |
| CNN-Based +HSFM +GMF (ours) | 0.0226 | 0.0759 | 0.0023 |

**Table 3 The performance of the stack@n of the interaction map as the input to our simple CNN-based proposed model.**

|  | Movielens | | Pinterest | |
| --- | --- | --- | --- | --- |
|  | HR@K | NDCG@K | HR@K | NDCG@K |
| Stack@1 | 0.9042 | 0.3221 | 0.9198 | 0.3276 |
| Stack@2 | 0.9563 | 0.3407 | 0.9390 | 0.3345 |
| Stack@3 | 0.9767 | 0.3479 | 0.9473 | 0.3374 |

**Note:**
$n \in \{1, 2, 3\}$.

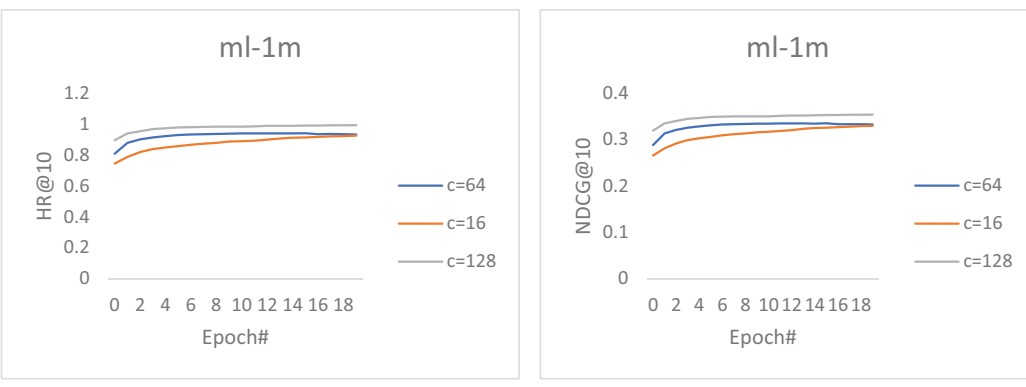

**Figure 2 The performance of our best model using different channel sizes on the MovieLens dataset.**

## Efficacy of stacking the interaction map (RQ2)

The stacking of the interaction map increased the latent signal, which allowed our proposed model to establish strong user-item relationships. From Table 3, the experiments demonstrated that the increase in the stacking of the interaction map has a proportionate impact on the effectiveness of the model performance. However, beyond stack@3, the model incurred overfitting, which impaired the model performance.

## Research of hyper-parameter (RQ3)

In this section, we investigated how the increased convolutional feature maps (channel size) impact the representation ability of our top-performing proposed model. As a result, we experimented with different channel sizes $c \in \{16, 64, 128\}$. Figure 2 shows the performance of our best model using different channel sizes on the movieLens dataset. We can observe that the increased number of channel sizes also improves the model's performance. However, these performance gains come at the expense of increased computational complexity and are time-consuming during training.

Furthermore, the charts show that all the curves increase steadily, and the feature map at channel 128 achieves the best performance. Moreover, channel 64 steadily outperformed channel 16 until the final epochs, where a slight difference in the convergence curve was noticed. These reflect our proposed model's strong expressiveness and generalization since

increasing the number of parameters adjust the model performance and does not lead to overfitting.

## CONCLUSIONS

In this article, we proposed a deep learning convolutional-based recommender system for modeling user-item correlation. The proposed method utilized convolution mechanisms for local-global feature selection and combined the generalized matrix factorization (GMF) to establish a more robust user and item relationship model that improved accuracy without overfitting. We conducted a series of experiments with two real-world datasets, and corresponding experimental results demonstrated that our proposed model has a higher recommendation accuracy and surpasses the baselines in the top-k recommendation task. In the future, we plan to explore other forms of deep learning techniques, such as transformers, to integrate better global user-item relationships beyond convolutional techniques.

### Funding

This work was funded by the Fund of National Natural Science Foundation of China (No. 61562010, 71964009), and the Research Projects of the Science and Technology Plan of Guizhou Province (No. [2021]449, No. [2021]261), No. [2023]010, No. [2023]276, No. [2023]338). The funders had no role in study design, data collection and analysis, decision to publish, or preparation of the manuscript.

### Grant Disclosures

The following grant information was disclosed by the authors:
Fund of National Natural Science Foundation of China: 62162010, 71964009.
Research Projects of the Science and Technology Plan of Guizhou Province: [2021]449, [2021]261, [2023]010, [2023]276, [2023]338.

### Competing Interests

The authors declare that they have no competing interests.

### Author Contributions

- Baboucarr Drammeh conceived and designed the experiments, performed the experiments, analyzed the data, performed the computation work, prepared figures and/or tables, authored or reviewed drafts of the article, and approved the final draft.
- Hui Li conceived and designed the experiments, performed the experiments, analyzed the data, performed the computation work, prepared figures and/or tables, authored or reviewed drafts of the article, and approved the final draft.

### Data Availability

 The original datasets used in this work are publicly available at:

- MovieLens datasets: GroupLens Research has collected and made available rating data sets from the MovieLens website (https://grouplens.org/datasets/movielens/). The data sets were collected over various periods of time, depending on the size of the set.

- The Pinterest dataset contains more than 1 million images associated with users who have "pinned" them. The full Pinterest dataset was released in the following article: Geng X, Zhang H, Bian J, Chia TS. 2015. Learning image and user features for recommendation in social networks. In: 2015 IEEE International Conference on Computer Vision (ICCV). Piscataway, IEEE: 4274–4282 DOI 10.1109/ICCV.2015.486.

The preprocessed dataset used in our research is formatted the same as the article (*He et al., 2017*) and is available at GitHub: https://github.com/Baboucar/HSFR/tree/master/HSFR/data.

The source code used in this work is available at GitHub and Zenodo: https://github.com/Baboucar/HSFR/tree/master/HSFR.

Baboucar. (2023). Baboucar/HSFR: Enhancing Neural Collaborative Filtering Using Hybrid Feature Selection (RecommenderSystem). Zenodo. https://zenodo.org/record/8027953.

## Supplemental Information

Supplemental information for this article can be found online at http://dx.doi.org/10.7717/peerj-cs.1456#supplemental-information.

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
