# Peer review of "Enhancing neural collaborative filtering using hybrid feature selection for recommendation"

_PeerJ Computer Science, doi:10.7717/peerj-cs.1456_

## Round 0.1 · original submission · Minor Revisions

Dear Authors,

Please revise and resubmit your manuscript. Thank you.

·

Basic reporting

The manuscript deals with an enhanced version of neural collaborative filtering for recommender systems which uses convolution instead of simple matrix factorization as a way to capture relationships between users and items. This brings an improved performance compared to previous implementations of neural CF.

The paper is clearly written, in a professional and academic language. The problem addressed in the paper is well motivated, and the bibliographic references are relevant and up to date too. The structure of the article is also clear, logical and easy to follow, and the graphics and figures are useful and informative. The datasets used in th study are public and accessible.

The proposed technique is clearly described, except for one point hat the authors should explain better: the embedding of users and items into features is not fully and completely specified. Since this is an important point for understanding the system design, the authors are encouraged to fix this part. The rest of the system is presented in a clear way.

Experimental design

The experimental result and the presentation of the results conforms to scholar standards and is well done: the datasets used for the experiments are significant and real, the performance metrics are clearly defined (and are typical in this field of study), and overall the numerical experiment have been created to find out the responses to the research questions posed in this paper. In this respect, the paper is absolutely correct.

This is, on the other hand, replicable research (provided the explanation of the embeddings pointed out previously is solved).

There is, however, some margin for improvements in the presentation:
1) The figures and performance plots do not have a high quality. It is suggested to improve them, both in quality and in clarity.
2) According to Figure 1, there seems to be no interaction in the computation between the user features and the item features. Please, explain this discrepancy.

Validity of the findings

3) The results with the proposed technique show also perfect prediction (roughly 99%) for HR, which is much better than with other methods. This sharp improvement should be explained and discussed better in the paper. Could there be some form of overfitting in the proposed system?

While the technique can be reproduced and tested again, such remarkable performance improvement must be explained directly by the authors, particularly the reasons for achieving those excellent results.

Reviewer 2 ·

Basic reporting

The research paper is well presented and well structured. The authors realized a new contribution in the domain of the collaborative filtering recommendation.
The authors should add some new references, the most updated one is 2020.
The results of the proposed model should be improved by its comparison with other models in terms of Within-Cluster Sum of Square, Silhouette coefficient, V-measure and Fowlkes-Mallows Scores

Experimental design

The proposed model is well presented and is useful to increase the understanding and the interpretation of the input features through the interaction maps and the correlation between items. Furthermore, the use of the Generalized Matrix Factorization a solution for preventing the overfitting issue.

Validity of the findings

The evaluation of the proposed model focused on the recommendation ranking using the hit ratio and the normalized discounted cumulative gain which are evaluation protocols in the recommendation domain. It will be more convenient if the authors show the efficiency of the model by measuring the Within-Cluster Sum of Square, Silhouette coefficient, V-measure and Fowlkes-Mallows Scores

Additional comments

Although the authors presented a link that contains the datasets, it will be better to present a paragraph to describe the features of each dataset.

---

## Round 0.2 · accepted · Accept

Dear Authors,

Your article is Accepted. While in production please improve the minor English language corrections. Thank you.

·

Basic reporting

The authors have addressed adequately my previous concerns on the experimental results. Once clarified, I consider this paper can be accepted for publication.

Experimental design

Additional and clear clarifications have been provided.

Validity of the findings

No further comments.

·

Basic reporting

The work presents a procedure for enhancing neural collaborative filtering using hybrid feature selection for a recommendation. The study is interesting and well-written. The coherence and general flow are fine, apart from occasional difficulty due to the use of a few complex sentences and limited syntax usage.

Experimental design

The methodological design is easy to follow and the mathematics are quite easy to follow. However, the algorithmic flowcharts are missing, but these do not impact the flow of the work.

Validity of the findings

The findings were fairly reported and the results support the conclusions.

Additional comments

One of the reviewers commented on the quality of the graphics. It appears that the graphs still suffer from poor presentation. However, it was quite easy to understand the results.